# The safety and efficacy of doxazosin in medical expulsion therapy for distal ureteric calculi: A meta-analysis

**Baozhong Yu[1‡], Xiang Zheng[1‡], Zejia Sun[1], Peng Cao[2], Jiandong Zhang[2], Zihao Gao[2], Haoyuan Cao[2], Feilong Zhang[2], Wei Wang[1]***

**1** Affiliated Beijing Chaoyang Hospital of Capital Medical University, Beijing, China, **2** Capital Medical University, Beijing, China

‡ These authors share first authorship on this work.
* weiwang0920@163.com

**Data Availability Statement:** All relevant data are within the paper.

**Funding:** The author(s) received no specific funding for this work.

## Abstract

### Purpose

Alpha-adrenergic receptor blockers can be effectively used in the context of medical expulsion therapy (MET) to treat ureteric stones. This study was designed to evaluate the safety and efficacy of doxazosin in MET relative to placebo or tamsulosin.

### Methods

We systematically searched the PubMed, the Cochrane Library, EMBASE, Chinese Academic Database, and Web of Science databases to select randomized controlled trials (RCT) that compared the use of doxazosin with placebo or tamsulosin to treat ureteric stones. All patients we included were limited to those diagnosed with visible stones in the distal ureter. The diameter of ureteric stones does not exceed 10 mm.

### Results

Eight trials comparing doxazosin with placebo or tamsulosin containing 667 patients were assessed in the final analysis. The meta-analysis showed that doxazosin effectively treated ureteric stones and was better than placebo in terms of efficacy. Relative to the placebo group, the expulsion rate of stones from the distal ureter (OR = 3.00, 95% CI [2.15, 4.19], I2 = 0%, P < 0.00001) was significantly increased, and the expulsion time (days) was shortened (mean difference) (MD) = −4.03, 95% CI [−4.53, −3.53], P < 0.00001). The doxazosin group experienced fewer pain episodes (MD = −0.78, CI = [−0.94, −0.23], I2 = 0%, P < 0.00001) than the placebo group. A subgroup analysis showed that the doxazosin group had a higher expulsion rate (of 5–10 mm stones) compared with the placebo group. Although doxazosin resulted in significantly more adverse effects compared with the placebo, the patient's symptoms were mild and no further medical interventions were required. Moreover, the expulsion time (days) was shorter for patients receiving doxazosin (MD = −0.61, 95% CI [−0.97, −0.24], $I^2$ = 39%, P = 0.001) than those receiving tamsulosin.

**Competing interests:** The authors have declared that no competing interests exist.

## Conclusion

Compared with the placebo group, patients receiving doxazosin had a greater expulsion rate, a reduced expulsion time, and fewer pain episodes. The expulsion time of doxazosin was shorter than that of tamsulosin.

## Background

Urinary calculi are frequent in urology and one of the most common causes of severe abdominal pain in the emergency department [1]. Ureteric calculi make up about 20% of cases of urolithiasis, and about 70% of the stones are "distal ureteric calculi (DUC)" or "lower ureteric stones (LUS)" found within the lower third of the ureter [2]. Treatment options for ureteric stones include spontaneous discharge, medical expulsion therapy (MET) [3], shock wave lithotripsy (SWL), ureteroscopic lithotripsy. Except waiting for observation and MET, other interventions have higher healthcare expenditures and are relatively invasive. Therefore, MET is usually the optimal treatment for patients with stones less than 10mm [4].

The drugs used for MET include alpha-blockers, calcium channel antagonists, phosphodiesterase inhibitors, and corticosteroids, which have all been shown to promote the discharge of ureteric stones. MET for ureteric stones has been used for some time, and the various drugs used are known to involve different expulsion mechanisms. The ureter has α1-adrenergic receptors, especially of subtype α1D, which are more numerous in the distal third of the ureter. They are important in lower ureteric physiology and affect the contraction of the detrusor and ureteric smooth muscle [5]. An increasing number of studies support the idea that α-adrenergic receptor blockers are effective treatments for ureteric stones [4, 6, 7].

Therefore, we conducted a meta-analysis to systematically evaluate the safety and effectiveness of doxazosin in the treatment of ureteric stones compared to placebo or tamsulosin, and hypothesized that doxazosin may have potential value in the treatment of ureteric calculi.

## Methods

### Study selection

Relevant studies were those meeting these inclusion criteria: (1) All patients were recruited into randomized controlled trial groups receiving either doxazosin or placebo/tamsulosin; (2) Patients were limited to those diagnosed by ultrasound and/or by radiology as having stones at the distal end of the ureter of a maximum dimension of 10 mm. Patients with fever, pregnancy, multiple stones in the kidneys and ureters, low blood pressure, severe heart disease, urinary tract infection, or a history of urological surgery, SWL, or endoscopic treatment were not included. Two authors (ZH Gao and HY Cao) independently reviewed all study titles and abstracts, and the full text was reviewed when necessary. Any discrepancies were resolved by discussion with X Zheng.

### Literature search

A comprehensive search for RCT was conducted based on the screening criteria. The target RCTs compared the efficacy and safety of doxazosin with those of placebo or tamsulosin in MET for ureteric stones. The PubMed, Cochrane Library, EMBASE, China Academic Database, and Web of Science databases were searched from the start of publication to August 2020. The search terms used in database were as follows: (urolithiasis or ureteric calculi or

ureteric calculi or renal calculi or colic or calculi* or calculi* or colic and ureter* or drug rejection therapy) and (doxazosin). We also manually searched journals related to urology, emergency medicine, and pharmacology to identify published trials and related review articles. We declare that there are no language restrictions when searching or retrieving documents.

### Data extraction and assessment of methodological quality

After the selected RCTs were included, two reviewers (ZH Gao and HY Cao) extracted information from each article. (1) The characteristics of the study were the first author, year of publication, total number of patients, patient sex, interventions, stone size, age, and follow-up time. (2) The primary outcomes were the expulsion rate and expulsion time, and the secondary outcomes were pain frequency and drug response. The subgroup analysis was conducted on the expulsion rate based on the size of the stones. (3) Methodological items were randomization, allocation concealment, participant and assessor blinding, incomplete outcome data, selective reporting of results, and publication bias. All the above information was cross-checked, and any differences were resolved via discussion.

### Analysis

The Review Manager (RevMan) software (v5.3.0) was used for statistical analyses. The $I^2$ statistics of the Mantel–Haenszel chi-squared test was employed to evaluate the heterogeneity. $I^2$ <50% was indicative of low heterogeneity, whereas values of >50% were considered to have high heterogeneity. When the $I^2$ value is less than 50%, a fixed-effect model was used; otherwise, we considered the heterogeneity and used a random effect model to provide a relatively conservative estimate. At the same time, sensitivity analysis was performed when the heterogeneity was high, and the combined effect size was reassessed by excluding the low-quality studies. An inverted funnel chart was used to assess the risk of publication bias. P<0.05 was the significance threshold.

## Results

### Study characteristics

Our search strategy yielded 667 citations, and we obtained 334 articles after excluding the duplicates. After the titles and abstract reviews passed the search criteria, 32 articles were obtained. The full texts of 32 articles were screened, and we included 714 patients from eight studies [8–15] (Fig 1) that evaluated the efficacy and safety of doxazosin treatment relative to a placebo or tamsulosin in MET. Of these, three studies [9, 12, 14] evaluated the efficacy of doxazosin and tamsulosin. Included study baseline characteristics are compiled in Table 1.

### Methodological quality

The results of the Cochrane risk assessment are shown in Fig 2, and the overall quality was good. Three trials [8, 11, 12] did not report hidden assignments, three trials [11–13] did not perform double-blinding during the process, and six trials [8–10, 14, 15] did not report blinding of participants and personnel. Funnel plots were symmetrical, consistent with a lack of publication bias associated with these eight studies (Fig 3).

### Doxazosin compared with placebo

**Expulsion rate.** In total, eight trials [8–15], which included 640 participants (320 and 320 in the doxazosin and placebo groups) (Fig 4A), were used to analyze the expulsion rates. Compared with that of the placebo, the expulsion rate of doxazosin was significantly higher

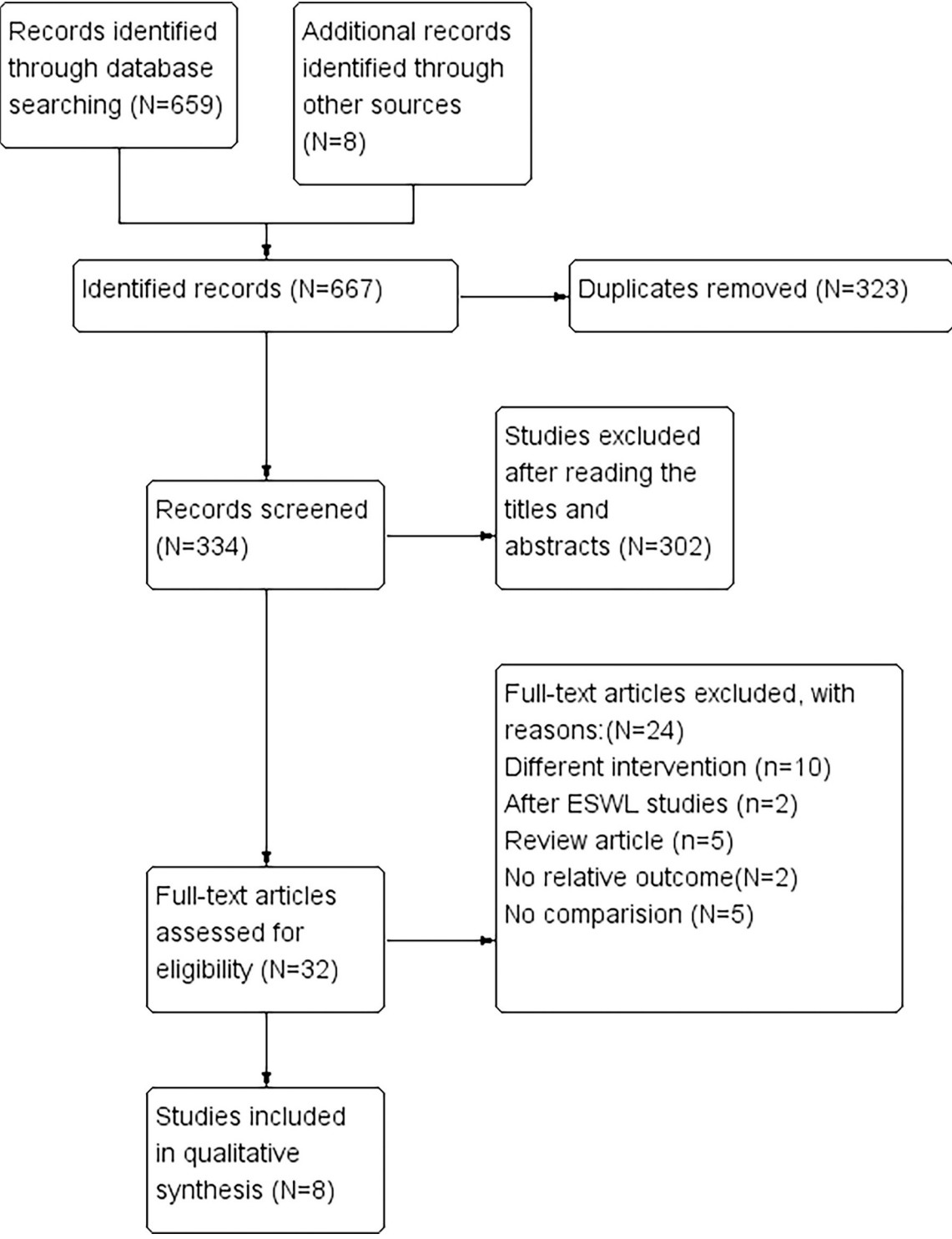

**Fig 1. Flowchart of literature selection process.** (D), (T)and (P) represent the doxazosin group, tamsulosin group and placebo group, respectively.

(OR = 3.00, 95% CI [2.15, 4.19], $I^2$ = 0%, P < 0.00001). Expulsion rates of ureteric stones in patients receiving doxazosin ranged from 64% to 81%, while those of the placebo were approximately 25% to 63%.

**Table 1. Study and patient characteristics.**

| Study | Therapy in experimental group | Therapy in control group | Country | Sample size | | Duration of treatment | Inclusion population |
|---|---|---|---|---|---|---|---|
| | | | | Experimental | Control | | |
| Zehri et al. | Doxazosin 2mg daily | Pakistan | Greece | D:33 | P:22 | 4weeks | Patients with ureteric stones in the distal ureter 4–7 mm in size. |
| Mshemish et al. | Doxazosin 4mg daily | Placebo Tamsulosin | Iraq | D:35 | P:35 T:33 | 45days | Patients aged ≥ 18 years with a single ureteric stone ≤10 mm size situated below the common iliac vessels |
| Sen et al. | Doxazosin 4mg daily | Placebo | Turkey | D:22 | P:19 | 3weeks | patients with symptomatic calculi and had unilateral distal ureteric stones of less than 10 mm in diameter |
| Yilmaz et al. | Doxazosin 4mg daily | Placebo Tamsulosin | Turkey | D:29 | P:28 T:29 | 4weeks | Aged ≥ 18 years with a single, unilateral, distal ureteric stone of 10 mm or smaller |
| Wei et al. | Doxazosin 4mg daily | Placebo | China | D:82 | P:80 | 2weeks | Patients middle or lower ureteral with a ureteric stones ≥4 mm and ≤10 mm |
| Zhou et al. | Doxazosin 4mg daily | Placebo | China | D:50 | P:50 | 2weeks | Patients with a ureteric stone of 5-10mm in size |
| Zhuo et al. | Doxazosin 4mg daily | Placebo Tamsulosin | China | D:32 | P:31 T:31 | 2weeks | Patients with ureteric stones of 4–10 mm |
| Liatsikos et al. | Doxazosin 4mg daily | Placebo | Greece | D:42 | P:31 | 4weeks | Patients unilateral, uncomplicated middle or lower ureteric stones which were ≤ 1 cm |

**Expulsion time.** Five RCTs [8, 9, 11, 12, 14] generated expulsion time data (Fig 4B), and there was heterogeneity ($I^2 = 70\%$) between the trials; however, the heterogeneity ($I^2 = 29\%$) was eliminated by omitting two low-quality trials. The results of the meta-analysis of random-effects models showed that doxazosin had shorter excretion days than placebo (MD = −4.48, 95% CI [−5.60, −3.36], P < 0.00001).

**Pain episodes.** Three trials [9, 11, 12], including 168 participants (86 and 82 in the doxazosin and placebo groups) (Fig 4C), resulted in data for patient pain episodes. Compared with placebo treatment, doxazosin treatment led to a statistically significant reduction in pain episodes (MD = −0.78, CI = [−0.94, −0.61], $I^2 = 0\%$, P < 0.00001).

**Drug adverse reaction.** A total of four trials [9, 11, 13, 15] compared the placebo group and the doxazosin group and produced data on drug adverse reactions (OR = 7.29; 95% CI = [1.63, 32.65]; $I^2 = 0\%$, P = 0.009) (Fig 4D). Doxazosin-related adverse events were generally mild, and most patients did not withdraw as a result of adverse reactions. The most common adverse events were nausea, headache, dizziness, backache, and postural hypotension.

## Expulsion rate (Stone size <5mm)

Two RCT reports [8, 9] provided data for the expulsion rates of stones of <5 mm in the treatment and placebo groups (Fig 5A). There were no statistically significant differences (OR = 2.23; 95% CI = [0.73, 6.76]; $I^2 = 0\%$, P = 0.16) between the expulsion rates of the treatment and placebo groups when the stones were less than 5 mm.

## Expulsion rate (of stones of 5–10 mm)

Data were reported for the expulsion rates of stones of 5–10 mm found in the placebo and doxazosin groups in two trials [8, 9] (Fig 5B). Relative to the placebo group, the doxazosin group had a higher expulsion rate for stones of 5–10 mm (OR = 5.11; 95% CI = [2.21, 11.82]; $I^2 = 0\%$, P = 0.0001).

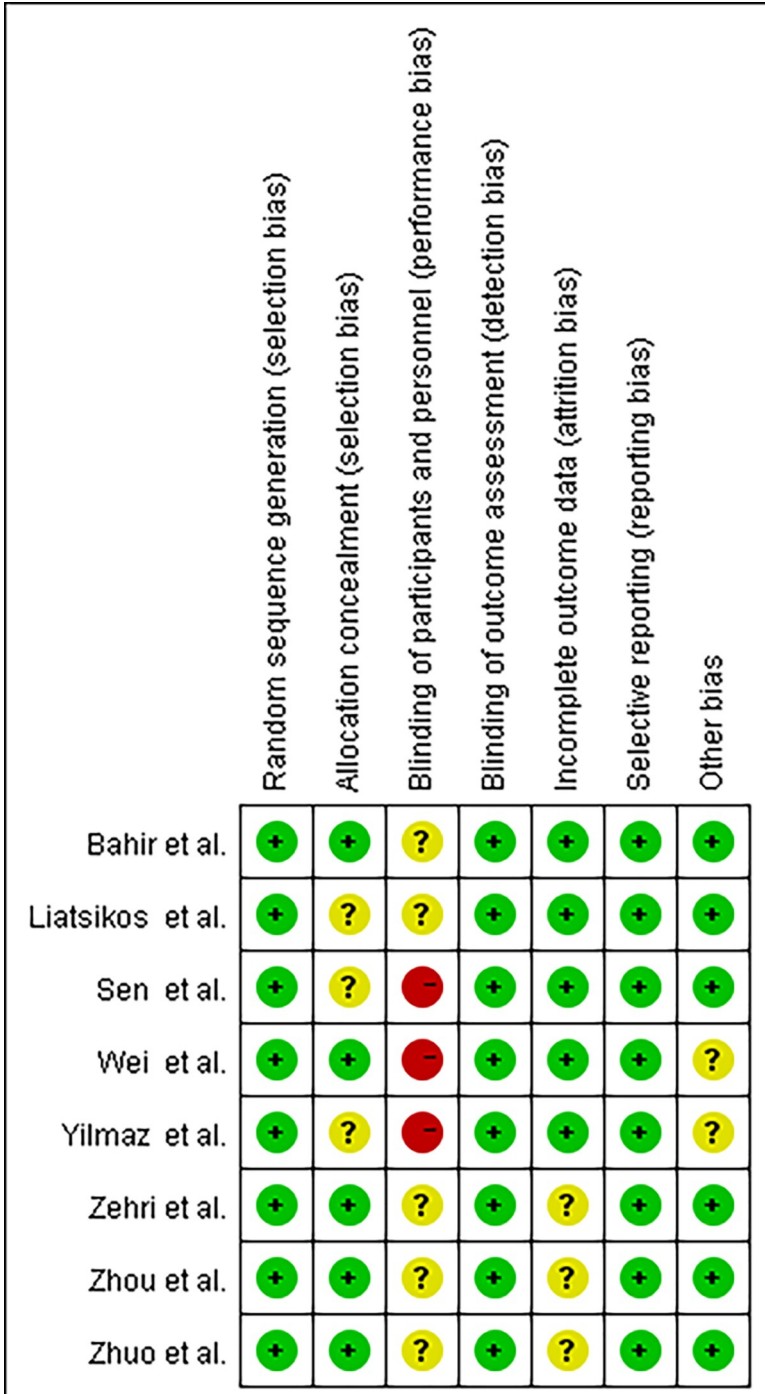

**Fig 2. Quality assessment graph.** (+) low risk, (?) unclear risk, (−) high risk.

## Doxazosin compared with tamsulosin

**Expulsion rate.** A total of three reports [9, 12, 14] on trials that included 187 participants (93 and 94 in the doxazosin and tamsulosin group) (Fig 6A) provided data for the expulsion rates. There were no statistically significant differences (OR = 0.82; 95% CI = [0.41, 1.64]; $I^2$ =

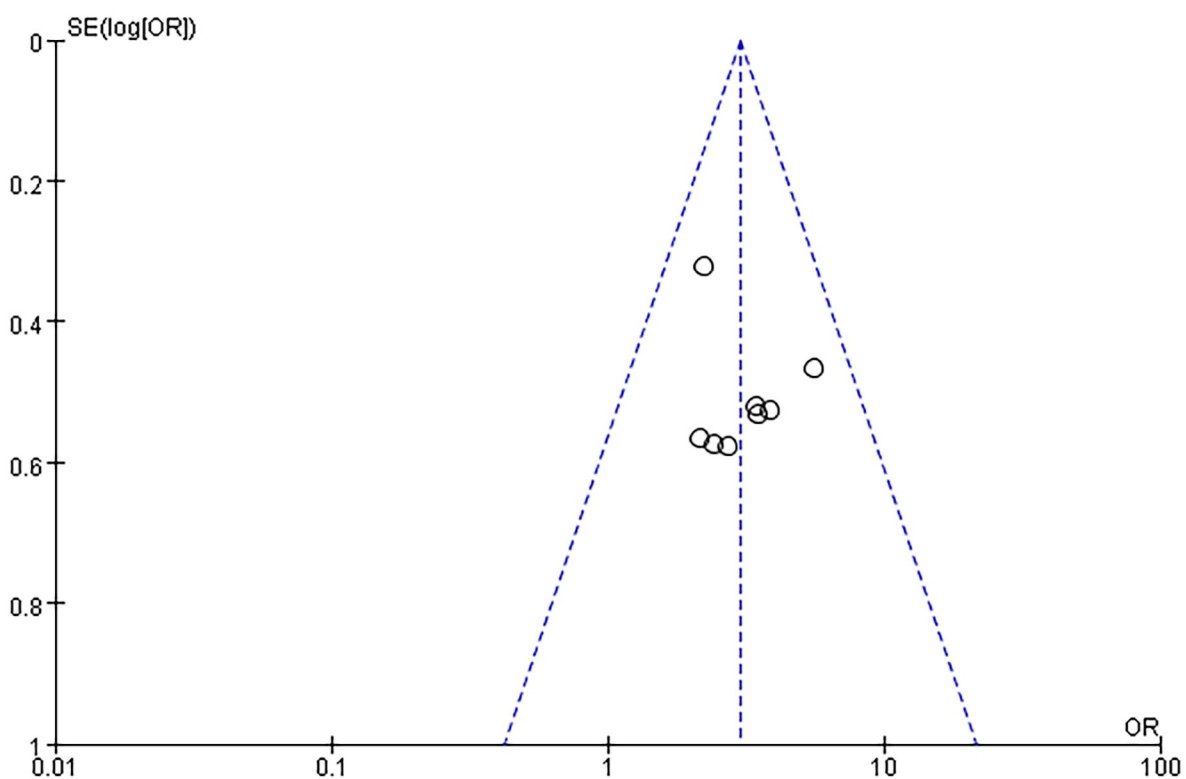

**Fig 3. Inverted funnel plot investigating publication bias.**

0%, P = 0.57) between the expulsion rates of the group treated with doxazosin compared with the group treated with tamsulosin.

**Expulsion time.** Expulsion time data were reported for three trials [9, 12, 14] that included 189 participants (96 in the doxazosin group and 93 in the doxazosin group) (Fig 6B). Compared with tamsulosin, doxazosin has shorter expulsion days (MD = −0.61, 95% CI [−0.97, −0.24], $I^2$ = 39%, P = 0.001).

**Pain episodes.** Two trials [9, 12] involving 126 participants (64 in the doxazosin group and 62 in the tamsulosin group) (Fig 6C) provided data for pain episodes. There was no significant difference (MD = −0.10; 95% CI = [−0.26, −0.05]; $I^2$ = 0%, P = 0.20) between the tamsulosin- and doxazosin-treated groups with regards to episodes of pain.

## Discussion

Urinary tract stones are very common. The prevalence is 1–19.1% in Asia, 5–9% in Europe, and 7–13% in North America[16, 17]. Although most small stones pass spontaneously, some do not, leading to infection or hydronephrosis. MET [18] has recently become the optimal treatment for patients with ureteric stones less than 10 mm and is considered both safe and effective. The mechanism of action of alpha-adrenergic blockers [19] involves the relaxation of the distal ureter by decreasing smooth muscle tension. Borghi et al. [20] showed that cases of spontaneous expulsion of stones through MET have been increasing, and various MET drugs have been studied [21, 22]. In 2017, Ye et al. [23] conducted a multi-center RCT in China and showed that tamsulosin is beneficial for the discharge of stones from the distal ureter. Furthermore, Hollingsworth et al. [24] conducted research on the therapeutic effects of all alpha-

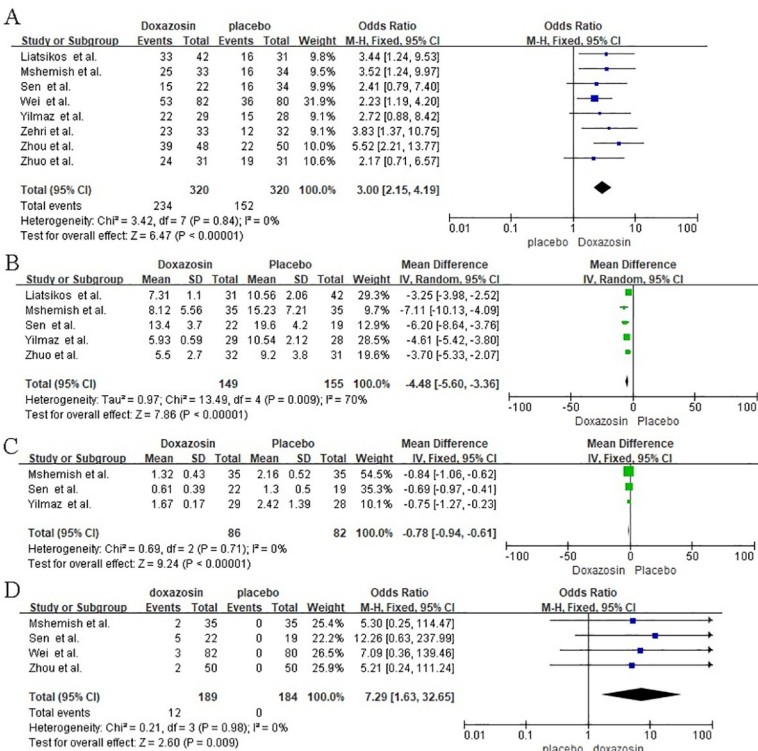

**Fig 4.** A: Meta-analysis results of expulsion rate between the groups. B: Meta-analysis results of expulsion time between the groups. C: Meta-analysis results of pain episodes between the groups. D: Meta-analysis results of drug adverse reaction between the groups.

blockers, and Skolarikos et al. [25] concluded that MET has a therapeutic effect for patients with ureteric stones < 10 mm. In summary, there is sufficient evidence to support the clinical application of MET in the treatment of ureteric stones.

Doxazosin is an adrenergic receptor antagonist. Because the number of published studies evaluating the effects of doxazosin on ureteric stones has increased, we conducted a meta-analysis to assess the effectiveness and safety of doxazosin. We included eight studies with 667 patients, and the results showed that, versus the placebo group, the doxazosin group showed an increased expulsion rate and a shortened expulsion time of distal ureteric stones. Subgroup

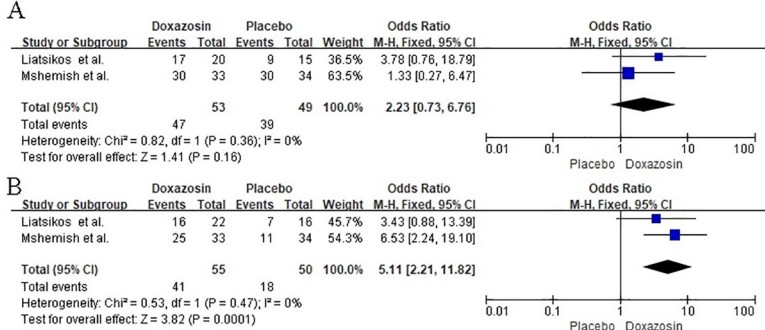

**Fig 5.** A: Meta-analysis results of expulsion rate (stones size <5mm) between the groups. B: Meta-analysis results of expulsion rate (stones size 5-10mm) between the groups.

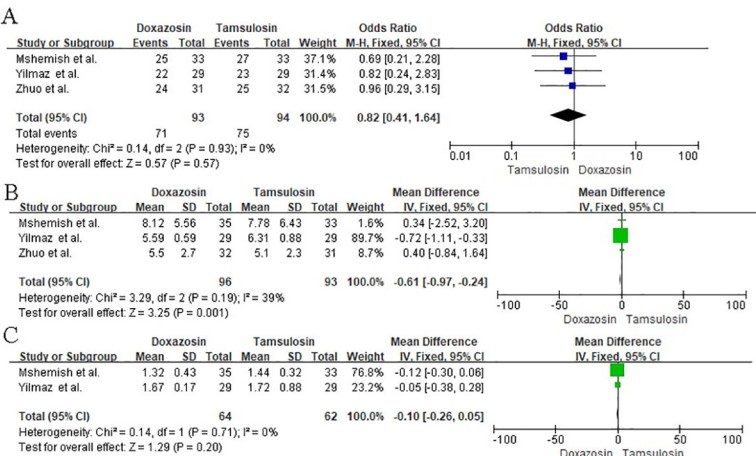

**Fig 6.** A: Meta-analysis results of expulsion rate between the groups. B: Meta-analysis results of expulsion time between the groups. C: Meta-analysis results of pain episodes between the groups.

analysis showed that doxazosin did not increase the rate of stone discharge in patients with stones of less than 5 mm; however, doxazosin significantly promoted the discharge of stones 5–10 mm in size. The rate of ureteric calculi excretion mainly depends on factors related to calculi [26], including size and location, and pathological factors such as urinary tract contractions. Coll et al. [27] revealed that ureteric stones less than 5 mm have a greater than 75% chance of spontaneous expulsion, whereas the spontaneous expulsion rates of ureteric stones of 5–7 mm, 7–9 mm, and >9 mm are 60%, 48%, and 25% respectively.

For larger stones (5~10 mm), most network meta-analyses focused on different medications for MET, and those findings suggested that α-blockers had the highest ranking for MET [28, 29]. The explanation is that α-adrenergic receptors are classified into three different subtypes of α-1A, α-1B, and α-1D, and the distribution in the human ureter is α-1D > α-1A > α-1B receptors[30]. Based on their findings, an α-1D-adrenoceptor blocker may provide better stone expulsion than an α-1A-adrenoceptor blocker. However, ureteral contractions were mainly mediated by α-1A-adrenoceptors in a hamster study[31]. Tsuzaka et al.[32] reported that an α-1A-adrenoceptor blocker provided more stone expulsions than an α-1D-adrenoceptor blocker. Sasaki et al. found that among α1-adrenoceptors, the α1A subtype played the major role in contraction in the human ureter[33]. Accordingly, α1A-adrenoceptor antagonists could become a useful medication for stone passage in urolithiasis patients. Doxazosin can bind to α1A adrenergic receptor and α1D adrenergic receptor[34]. Therefore, doxazosin may be a useful drug for patients with ureteric stones.

In addition, our analysis of the results showed doxazosin to be associated with adverse reactions. Regarding the adverse reactions reported in the included studies, there were 12 patients in the doxazosin group with side effects. Minor adverse reactions mainly include dizziness, nausea, and orthostatic hypotension, and, generally, no further medical intervention is required. Because the use of doxazosin in MET has certain adverse effects, we still need to pay attention to the patient's physical condition even though medical intervention is unnecessary in most cases.

Many studies have proved that MET is indeed effective as a means of treating patients with stones < 10 mm in the distal ureter. We can conclude that doxazosin produces a higher expulsion rate, faster expulsion time, and fewer pain episodes compared with placebo treatment. Moreover, the expulsion time of the doxazosin group was faster than the tamsulosin group.

Hence, we concluded that doxazosin has potential as a MET for ureteric stones, and we aim to further study the effects of doxazosin on patients with ureteric stones. The use of doxazosin may provide patients with an additional drug option for promoting stone removal without highly invasive and expensive interventions [35].

Our meta-analysis had some limitations. Research quality estimates are affected by insufficient amounts of information provided in publications or methodological differences between these studies. Some articles [10, 12] did not report the incidence of any minor or severe drug-related adverse reactions, and we did not obtain relevant data involving deviations. Unfortunately, one study [10] failed to detail the standard deviation of the expulsion time; therefore, the study quality estimation was influenced by the inadequate information provided in the publications or the methodological differences among the included studies. Furthermore, we were unable to include unpublished study data, potentially biasing our results. Because of the limited number of studies, the long-term safety and effectiveness of doxazosin could not be inferred. Despite a comprehensive search, positive studies seem to be easier to publish and publication bias may exist. Therefore, more high-quality tests with larger samples are needed to increase our understanding of the effectiveness and safety of doxazosin for the treatment of ureteric stones.

## Conclusion

This meta-analysis showed that doxazosin is more effective for the treatment of distal ureteric stones and pain control than a placebo. Furthermore, doxazosin has a shorter stone expulsion time than tamsulosin. We concluded that doxazosin can be used as a MET for ureteric stones.

## Author Contributions

**Conceptualization:** Zihao Gao.

**Data curation:** Xiang Zheng, Zihao Gao.

**Software:** Xiang Zheng, Jiandong Zhang, Haoyuan Cao, Feilong Zhang.

**Supervision:** Zejia Sun, Peng Cao, Wei Wang.

**Writing – original draft:** Baozhong Yu, Xiang Zheng.

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
