## [Decision Letter · Decision Letter 0]

28 Oct 2020

PONE-D-20-25593

The safety and efficacy of doxazosin in medical expulsion therapy for distal ureteral calculi: a meta-analysis

PLOS ONE

Dear Dr. Wang,

Thank you for submitting your manuscript to PLOS ONE. After careful consideration, we feel that it has merit but does not fully meet PLOS ONE’s publication criteria as it currently stands. Therefore, we invite you to submit a revised version of the manuscript that addresses the points raised during the review process.   The unedited reviewers comments are below

We look forward to receiving your revised manuscript.

Kind regards,

Adrian Stuart Wagg, MD

Academic Editor

PLOS ONE

Journal Requirements:

2. Please confirm that you have included all items recommended in the PRISMA checklist including the full electronic search strategy used to identify studies with all search terms and limits for at least one database.

3. Please include your tables as part of your main manuscript and remove the individual files. Please note that supplementary tables should be uploaded as separate "supporting information" files.

Reviewers' comments:

Reviewer's Responses to Questions

**Comments to the Author**

1. Is the manuscript technically sound, and do the data support the conclusions?

Reviewer #1: Yes

Reviewer #2: Yes

2. Has the statistical analysis been performed appropriately and rigorously? 

Reviewer #1: Yes

Reviewer #2: Yes

3. Have the authors made all data underlying the findings in their manuscript fully available?

Reviewer #1: Yes

Reviewer #2: Yes

4. Is the manuscript presented in an intelligible fashion and written in standard English?

Reviewer #1: Yes

Reviewer #2: No

5. Review Comments to the Author

Reviewer #1: Reviewers report: the safety and efficacy of doxazosin in medical expulsion therapy for distal ureteral calculi: a meta analysis

The objective of the study was the evaluation of the safety and efficacy of doxazosin relative to placebo or temp solution in medical expulsion therapy for distal ureteral calculi.

Abstract: This is an accurate summary of the following study

introduction: There are some minor grammatical alterations needed in the introduction but these are editorial and do not necessarily need alterations by the authors.

The size of stones should be specified do the authors mean 10 millimeters in diameter or length for example?

The last sentence of the background can be deleted this study does not assess an hypothesis but merely analyzes the results of previously tested hypotheses

Literature search:

I think it would be worthwhile changing the flow of the sections here. the inclusion and exclusion criteria should perhaps precede the literature search. Were there any language restrictions placed on the searches?

Study selection: although not required by the PRISMA reporting system could the authors tell us how many articles required discussion following discrepancy after independent review?

I assume that patient sex rather than gender was extracted?

Did the authors use a standard method to assess the quality of the study which was the basis of information in figure two?

The statistical analysis is appropriate to the methodology

results: In the sentence following the notes on table one the letters DP&T are incorrectly ordered in terms of explaining the definitions

I think the correct description is ureteric rather than ureteral stones. this needs correcting throughout

The unit of expulsion time is missing, I assume this is hours but this needs stating

Could the authors give the proportions of exposed patients who experienced each of the listed adverse events?

Are there data for comparative rates of adverse events between doxazosin and tamsulosin?

Discussion: The sentence following the paragraph “we found differences in the risk of common adverse reactions…” appears to be redundant . Were the authors able to ascertain whether any medical intervention was required for drug associated adverse events?

Where the authors use the term eviction time, I think they mean expulsion time

The authors rightly note that they were unable to include unpublished study data . Did they perhaps investigate whether there were such data ? For example, a search of a clinical trials registry might have uncovered other studies of relevance for which there were no results. In terms of limitations, were included studies English language only? This needs clarification

If the authors used the meta analytical results to conduct a power and sample size calculation , what number of patients that would be required for a fully powered study? They conclude that more high quality tests with larger samples might be needed but this may not be the case

Reviewer #2: Thank you for asking me to review this meta-analysis of doxazocin for the treatment of ureteric stones.

Overall this is a valid subject and the meta analysis is conducted well. The inclusion and exclusion criteria for the papers are well described and the statistical methods seem appropriate.

I do have a few comments that should be addressed.

The word "ureteral" is used throughout. Would "ureteric" be a better choice?

Abstract:

There is an inconsistency regarding the episodes of pain between groups, with the doxazocin group experiencing "more episodes of pain" (results, line 7) and "fewer pain episodes" (conclusion). This should be corrected.

Background:

"There are long waiting times for observation and MET". This sentence seems counterintuitive - are the authors suggesting that these treatments take a long time to work, or that there are long wait times to commence treatment, which would be the usual interpretation of this wording. This should be clarified.

Data extraction and assessment of methodological quality

"Patient genders" refers to biological sex, not gender. This should be corrected.

Discussion

The claim that 1 to 2% of hospital admissions are due to ureteral stones should be referenced or removed.

"70% of ureteral stones develop within the lower part of the ureter". Is this correct? surely stones form in the kidney and then move to the ureter? Either way it needs referencing.

The discussion section is overly long and repeats much of the method and background. It should be heavily edited or rewritten to focus on discussion the findings of the work, rather than reahashing previous sections.

Limitations; the limitation of the study not reporting their SD is a limitation of the statistical method, not of the software. The sentence referring to "Review Manager" should be edited.

---

## [Author Response · Author response to Decision Letter 0]

15 Nov 2020

Nov, 2020 15.

RE: manuscript reference number: PONE-D-20-25593.

Title: The safety and efficacy of doxazosin in medical expulsion therapy for distal ureteral calculi: a meta-analysis.

Dear Editors and Reviewers:

Thank you for your comments regarding the submitted paper, which will help us improve it to a better scientific level. We have made detailed revisions in accordance with the reviewer ’s comments. The amendments are highlighted in red in the revised manuscript. Point by point responses to the Reviewers’ comments are listed in the following letter.

I am looking forward to hearing from you.

Sincerely,

Baozhong Yu

Department of Urology

Beijing Chaoyang Hospital, Capital Medical University

8 Gong Ti Nan Road, Chaoyang District, Beijing, 100020, China

Reply to Reviewer：

Reviewer1: Thank you for your reviewer’s comments concerning our study by Wang and colleagues. We are very grateful for your valuable comments on the review. The following is my response to each question.

Question1: The size of stones should be specified do the authors mean 10 millimeters in diameter or length for example?

Answer1: Thank you very much for Reviewer’s suggestion. We re-modified the summary section and marked the size of the stones in the method section.

Question2: The last sentence of the background can be deleted this study does not assess a hypothesis but merely analyzes the results of previously tested hypotheses.

Answer2: Thanks for your suggestion. We also think this sentence is inappropriate. So, we re-edited this sentence.

Question3: I think it would be worthwhile changing the flow of the sections here. the inclusion and exclusion criteria should perhaps precede the literature search. Were there any language restrictions placed on the searches?

Answer3: Thank you very much for Reviewer’s suggestion. We revised the article and put the inclusion and exclusion criteria in front of the literature search. We do not have any language restrictions in the process of searching documents.

Question4: although not required by the PRISMA reporting system could the authors tell us how many articles required discussion following discrepancy after independent review?

Answer4: Thank you very much for Reviewer’s suggestion. After we independently reviewed the literature, there are a total of 5 documents that need to be discussed.

The literature was excluded considering that the intervention measures did not meet or the patients were not randomly assigned.

Question5: I assume that patient sex rather than gender was extracted?

Answer5: Thank you very much for Reviewer’s suggestion. We noticed that our words were inappropriate. We modified gender to sex. 

Question6: Did the authors use a standard method to assess the quality of the study which was the basis of information in figure two?

Answer6: Thank you very much for Reviewer’s suggestion. We evaluated each included study according to the standard of Cochrane's risk assessment tool.

Question7: In the sentence following the notes on table one the letters DP&T are incorrectly ordered in terms of explaining the definitions.

Answer7: Thank you for your reminder. In response to the reviewers' comments, we have revised the note and adjusted the order of letters DP&T.

Question8: I think the correct description is ureteric rather than ureteral stones. this need correcting throughout

Answer8: Thanks for your suggestion. We noticed that our usage was inappropriate. We checked the article and modified ureteral stone to ureteric stone. 

Question9: The unit of expulsion time is missing; I assume this is hours but this need stating.

Answer9: Thank you for your reminder. We state that the unit of expulsion time is hours.

Question10: Could the authors give the proportions of exposed patients who experienced each of the listed adverse events?

Answer10: Thanks for your suggestion. In the four studies, there were 212 people in the doxazosin group, of which 7 patients had hypotension, 7 patients had nausea and vomiting, and 5 patients had dizziness.

Question11: Are there data for comparative rates of adverse events between doxazosin and tamsulosin?

Answer11: Thank you very much for Reviewer’s suggestion. Four of the eight included studies compared doxazosin and tamsulosin. But only one of the four studies reported adverse drug reactions. Insufficient data are difficult to compare the incidence of drug adverse reaction between doxazosin and tamsulosin.

Question12: The sentence following the paragraph “we found differences in the risk of common adverse reactions…” appears to be redundant. Were the authors able to ascertain whether any medical intervention was required for drug associated adverse events?

Answer12: Thanks for your suggestion. The sentence following the paragraph “we found differences in the risk of common adverse reactions…” appears to be inappropriate. So, we deleted and revised this sentence. According to the included studies, the symptoms of patients with adverse reactions can generally be relieved. However, it is difficult to determine that medical intervention is not required after adverse events. Therefore, we believed that the sentence in the article is inappropriate, so we deleted this sentence.

Question13: Where the authors use the term eviction time, I think they mean expulsion time

Answer13: Thank you for your reminder. We noticed that our words were inappropriate. We modified eviction time to expulsion time.

Question14: The authors rightly note that they were unable to include unpublished study data. Did they perhaps investigate whether there were such data? For example, a search of a clinical trials registry might have uncovered other studies of relevance for which there were no results. 

Answer14: Thank you very much for Reviewer’s suggestion. Considering that some researchers missed the registration time due to negligence and failed to upload the research and related data to the website. We took this into consideration and said that we they were unable to include unpublished study data. But it is unreasonable to write this without evidence, we deleted this sentence.

Question15: In terms of limitations, were included studies English language only? This needs clarification

Answer15: Thanks for your suggestion. We clarified that the research includes not only the English language, but also other languages.

Reviewer2: Thank you for your reviewer’s comments concerning our study by Wang and colleagues. We are very grateful for your valuable comments on the review. The following is my response to each question.

Question1: The word "ureteral" is used throughout. Would "ureteric" be a better choice?

Answer1: Thanks for your suggestion. We noticed that our usage was inappropriate. We checked the article and modified ureteral stone to ureteric stone. 

Question2: There is an inconsistency regarding the episodes of pain between groups, with the doxazocin group experiencing "more episodes of pain" (results, line 7) and "fewer pain episodes" (conclusion). This should be corrected.

Answer2: Thanks for your suggestion. We checked the entire article and noticed inconsistencies in the article, so we revised this part again.

Question3: "There are long waiting times for observation and MET". This sentence seems counterintuitive - are the authors suggesting that these treatments take a long time to work, or that there are long wait times to commence treatment, which would be the usual interpretation of this wording. This should be clarified.

Answer3: Thanks for your suggestion. We realize that there is a problem with the wording of this sentence. What we want to clarify is that in addition to conservative treatment and MET, other interventions are high and invasive. We checked and revised this sentence.

Question4: "Patient genders" refers to biological sex, not gender. This should be corrected.

Answer4: Thank you very much for Reviewer’s suggestion. We noticed that our words were inappropriate. We modified gender to sex.

Question5: The claim that 1 to 2% of hospital admissions are due to ureteral stones should be referenced or removed. "70% of ureteral stones develop within the lower part of the ureter". Is this correct? surely stones form in the kidney and then move to the ureter? Either way it needs referencing.

Answer5: Thank you very much for Reviewer’s suggestion. We noticed that there was a problem with this part of the statement, so we revised this sentence again.

Question6: The discussion section is overly long and repeats much of the method and background. It should be heavily edited or rewritten to focus on discussion the findings of the work, rather than reahashing previous sections.

Answer6: Thank you very much for Reviewer’s suggestion. We realized that the discussion section was too long and repeated many methods and backgrounds. So, we re-edited or rewritten the discussion part.

Question7: Limitations; the limitation of the study not reporting their SD is a limitation of the statistical method, not of the software. The sentence referring to "Review Manager" should be edited.

Answer7: Thanks for your suggestion, we realize that this sentence is wrong. We checked and revised this sentence.

Sincerely,

Baozhong Yu

---

## [Decision Letter · Decision Letter 1]

15 Dec 2020

PONE-D-20-25593R1

The safety and efficacy of doxazosin in medical expulsion therapy for distal ureteric calculi: a meta-analysis

PLOS ONE

Dear Dr. Wang,

Thank you for submitting your manuscript to PLOS ONE. After careful consideration, we feel that it has merit and there are only some minor considerations suggested by the reviewers, these are below. Therefore, we invite you to submit a revised version of the manuscript that addresses the points raised during the review process.

We look forward to receiving your revised manuscript.

Kind regards,

Adrian Stuart Wagg, MD

Academic Editor

PLOS ONE

Reviewer #1: Some minor comments

Thank you for asking me to review a revised version of this paper. I have reviewed the previous comments In response to which the authors have made considerable revision to their paper.

My comments on the revised version are as follows:

In the final sentence of the introductory section, the authors might replace the words “which may find “ with “and hypothesized that “

In the second line of the study selection section, the word randomized is repeated In the same sentence. Perhaps the authors might replace the 1st “randomized" with either recruited or enrolled ?

In the literature search section, the authors should state that there were no language limitations on their search or retrieval

In the expulsion time section the authors omit the units of time measured in their text

In the pain episodes section, perhaps “compared with placebo treatment, doxazosin treatment led to a statistically significant reduction in pain episodes”?

In the drug adverse reaction in section, in the text the authors should specify the comparator where they quote the odds ratio

In the expulsion time section when comparing doxazosin to tamsulosin, the authors repeat doxazsin in each group and the unit of time is missing

Reviewer #2: Thank you for the opportunity to review this revised manuscript. I note that the comments of myself and my co-reviewer have been addressed in full.

I have three minor points to consider.

Methods: Analysis. "When the I2 value suggested very low heterogenicity" - the cut off for "very low" is not given and should be added.

Discussion: "The mechanism of action of alpha-adrenergic blockers [23] involves the relaxation of the distal ureter by decreasing smooth muscle tension instead of eliminating its activity" this sentence is unclear and should be reworded. The following sentence "In 2005, Sigala et al. [22] determined that the distal ureter expresses higher levels of α1-adrenoceptor mRNA than the proximal and medial ureters." doesn't really add much and could be removed.

In the conclusions you state "We concluded that doxazosin has potential as a MET for ureteric stones." It seems that your work demonstrates more than this, and the conclusion could be firmer.

---

## [Author Response · Author response to Decision Letter 1]

17 Dec 2020

Reviewer1: Thank you for your reviewer’s comments concerning our study by Wang and colleagues. We are very grateful for your valuable comments on the review. The following is my response to each question.

Question1: In the final sentence of the introductory section, the authors might replace the words “which may find “with “and hypothesized that “

Answer1: Thank you very much for Reviewer’s suggestion. We noticed that our words were inappropriate. We modified “which may find” to “and hypothesized that”.

Question2: In the second line of the study selection section, the word randomized is repeated in the same sentence. Perhaps the authors might replace the 1st “randomized" with either recruited or enrolled?

Answer2: Thanks for your suggestion. We noticed that our words were inappropriate. We modified the 1st “randomized” to “recruited”.

Question3: In the literature search section, the authors should state that there were no language limitations on their search or retrieval

Answer3: Thank you very much for Reviewer’s suggestion. We revised the article and declare that there are no language restrictions when searching or retrieving documents.

Question4: In the expulsion time section the authors omit the units of time measured in their text

Answer4: Thank you very much for Reviewer’s suggestion. In the expulsion time section, the unit of time measured in the text is days. And reflect the time unit in the text

Question5: In the pain episodes section, perhaps “compared with placebo treatment, doxazosin treatment led to a statistically significant reduction in pain episodes”?

Answer5: Thank you very much for Reviewer’s suggestion. We noticed that this sentence was inappropriate. As the reviewer said, what we want to express is that the doxazosin group can reduce the frequency of pain episodes compared with the placebo group. We checked and revised this sentence. 

Question6: In the drug adverse reaction in section, in the text the authors should specify the comparator where they quote the odds ratio

Answer6: Thank you very much for Reviewer’s suggestion. We reviewed this part and made it clear that the comparators were the placebo group and the doxazosin group. We checked and revised this part.

Question7: In the expulsion time section when comparing doxazosin to tamsulosin, the authors repeat doxazsin in each group and the unit of time is missing

Answer7: Thank you for your reminder. In the expulsion time section, the unit of time measured in the text is days. And reflect the time unit in the text.

Reviewer2: Thank you for your reviewer’s comments concerning our study by Wang and colleagues. We are very grateful for your valuable comments on the review. The following is my response to each question.

Question1: Methods: Analysis. "When the I2 value suggested very low heterogenicity" - the cut off for "very low" is not given and should be added.

Answer1: Thanks for your suggestion. I2 <50% indicates low heterogeneity. When the I2 value is less than 50%, the fixed effects model is used. We checked and revised this sentence.

Question2: Discussion: "The mechanism of action of alpha-adrenergic blockers [23] involves the relaxation of the distal ureter by decreasing smooth muscle tension instead of eliminating its activity" this sentence is unclear and should be reworded.

Answer2: Thanks for your suggestion. We noticed that the sentence was unclear, so we revised it again to make it more concise and clearer.

Question3: The following sentence "In 2005, Sigala et al. [22] determined that the distal ureter expresses higher levels of α1-adrenoceptor mRNA than the proximal and medial ureters." doesn't really add much and could be removed.

Answer3: Thanks for your suggestion. We realized that this sentence doesn't really add much. So, we removed this sentence

Question4: In the conclusions you state "We concluded that doxazosin has potential as a MET for ureteric stones." It seems that your work demonstrates more than this, and the conclusion could be firmer.

Answer4: Thank you very much for Reviewer’s suggestion. We have revised the conclusion part. Make the conclusion firmer.

---

## [Editor Report · Decision Letter 2]

7 Jan 2021

The safety and efficacy of doxazosin in medical expulsion therapy for distal ureteric calculi: a meta-analysis

PONE-D-20-25593R2

Dear Dr. Wang,

We’re pleased to inform you that your manuscript is now suitable for publication following academic editor review of the minor amendments and will be formally accepted for publication once it meets all outstanding technical requirements.

Kind regards,

Adrian Stuart Wagg, MD

Academic Editor

PLOS ONE

---

## [Editor Report · Acceptance letter]

13 Jan 2021

PONE-D-20-25593R2 

The safety and efficacy of doxazosin in medical expulsion therapy for distal ureteric calculi: a meta-analysis

Dear Dr. Wang:

I'm pleased to inform you that your manuscript has been deemed suitable for publication in PLOS ONE. Congratulations! Your manuscript is now with our production department. 

Kind regards, 

on behalf of

Dr. Adrian Stuart Wagg 

Academic Editor

PLOS ONE